# Effect of Glycosidase Production by *Rhodotorula mucilaginosa* on the Release of Flavor Compounds in Fermented White Radish

**DOI:** 10.3390/foods14071263

**Published:** 2025-04-03

**Authors:** Huixin Zhang, Rui Wang, Yaoying Wang, Yanfei Wang, Tao Wang, Chuanqi Chu, Shengbao Cai, Junjie Yi, Zhijia Liu

**Affiliations:** 1Faculty of Food Science and Engineering, Kunming University of Science and Technology, Kunming 650500, China; z15829158446@163.com (H.Z.); wwwr26@163.com (R.W.); 15798951286@163.com (Y.W.); wtao_2017@163.com (T.W.); chuanqichu@aliyun.com (C.C.); caikmust2013@kmust.edu.cn (S.C.); junjieyi@kust.edu.cn (J.Y.); 2Yunnan Key Laboratory of Plateau Food Advanced Manufacturing, Kunming 650500, China; 3International Green Food Processing Research and Development Center of Kunming City, Kunming 650500, China

**Keywords:** *Rhodotorula mucilaginosa*, fermented vegetables, volatile compounds, glycosidase, flavor

## Abstract

Fermented vegetables are highly valued by consumers for their distinct flavors and rich nutritional content. Microbial fermentation imparts distinct flavors to these vegetables, with red yeast being a common microorganism involved in the fermentation process. However, studies on the impact of red yeast on flavor development in fermented vegetables remain scarce. This study employed multi-omics to analyze the effect of glycosidase produced by *Rhodotorula mucilaginosa* on the release of bound flavor compounds in vegetables. The results indicate that the yeast possesses multiple glycosidase-encoding genes, with the activities of α-galactosidase, β-glucosidase, and α-mannosidase being detected. Following the inoculation of yeast into fermented vegetable juice, a significant increase was observed in the expression of the β-glucosidase gene (*bglX*) and the α-glucosidase maltase gene (*malL*), alongside an increase in the content of flavor compounds correlated with the enzymatic activity detected. The application of commercial glycosidase to vegetable juice resulted in increased levels of cis-2-pentenol, hyacinthin, geranylacetone, and 1-dodecanol, consistent with findings from yeast-fermented vegetable juice. Thus, *Rhodotorula mucilaginosa* can secrete glycosidases that hydrolyze and release endogenous bound flavor compounds in vegetables, thereby enhancing the flavor quality of the final product.

## 1. Introduction

Fermented vegetables are traditional foods that extend the shelf life of vegetables and are appreciated for their unique flavors. Vegetables such as napa cabbage, cabbage, cauliflower, and radish are commonly used as raw materials in fermented vegetable production. Many studies have shown that fermented vegetables can provide nutrients including vitamins, proteins, carbohydrates, and antioxidants [1,2,3]. Moreover, fermented vegetables are an important method in the intensive processing of vegetables, possessing substantial market values.

Fermented vegetables contain diverse taste-active compounds, which are significant factors contributing to consumers’ preference. The formation of these flavor substances mainly arises from the metabolic activities of microorganisms during the fermentation process of vegetables. Microorganisms can utilize the nutritional components present in vegetables to synthesize certain flavor compounds [4,5]. Furthermore, they can modify and transform the original flavor components of the vegetables by producing various enzymes, or release the bound flavor components in the vegetables through enzymatic hydrolysis [6,7]. These include glycosidically bound compounds, such as geranyl-β-D-glucoside and cyanogen glycosides, which are water-soluble and odorless in nature, and can be converted into geraniol [8] and α-terpineol [9] after hydrolysis. These compounds have been considered as the core flavors in fermented tea [10] and wine [11].

Many microorganisms involved in vegetable fermentation can produce glycosidases to release glycoside-bonded flavor substances. Previous studies have isolated and purified one strain of *Lentilactobacillus buchneri* with β-glucosidase activity from traditionally fermented vegetables in Korea [12]. The results of this study showed that fermentation by β-glucosidase-producing lactic acid bacteria is beneficial to the conversion of bound phenols and increases the total phenolic content. It was identified that yeasts are capable of producing β-D-glucosidase, α-L-rhamnosidase, β-D-xylosidase, and α-L-arabinosidase to enhance the aromatic profile of dry white wines [13].

Among these microorganisms, red yeast *Rhodotorula* sp. are interesting microorganisms. *Saccharomyces cerevisiae* primarily relies on fermentation, resulting in the production of ethanol, organic acids, and carbon dioxide [14]. In environments with a low pH and high ethanol concentration, glycosidase activity is suppressed, making it difficult to hydrolyze bound aroma precursors. Consequently, the main flavor contributions come from fermentation byproducts, such as esters and alcohols [15]. In contrast, *Rhodotorula* spp. utilize oxidative metabolism, which predominantly produces carbon dioxide, water, and ATP. They can maintain activity during the whole fermentation process [16]. During the fermentation process, it can metabolize various types of sugars and produce volatile compounds such as esters and alcohols, significantly enhancing the flavor of fermented vegetables [17,18]. Research has shown that it can also synthesize antioxidant substances like β-carotene during fermentation [19,20], thereby improving the nutritional value of the product. Our previous study also found that the main role of red yeast in fermenting vegetables is to convert and release the original flavor components in vegetables [21]. However, which glycosidases *Rhodotorula* spp. can produce and which bound flavor components can be released still need further verification.

Therefore, based on previous studies and the high juice yield of radishes, the relevant glycosidase-encoding genes were initially identified in *R. mucilaginosa* (I2002) by whole-genome sequencing analysis and the glycosidase activity was verified in the YPD medium and radish juice fermented by this yeast. To further confirm the active genes and enzymes for releasing bound aroma compounds, transcriptomics and proteomics of strains were analyzed. Then, *R. mucilaginosa* and commercial glycosidase were used to treat radish juice. Common volatile compounds were identified by these two treatments and considered as the aromas released by glycosidases produced by *R. mucilaginosa*. This study provides insights into the formation mechanism of flavor compounds in fermented vegetables and offers a theoretical basis for improving the quality of fermented vegetables.

## 2. Materials and Methods

### 2.1. Genomics Analysis

The selected red yeast *R. mucilaginosa* (I2002) was isolated from fermented vegetables obtained in Yunnan Province in China and preserved in the culture collection of Yunnan Key Laboratory of Plateau Food Advanced Manufacturing. The strain was retrieved from the glycerol tubes stored at −80 °C and allowed to thaw at room temperature. The strain solution was inoculated with a sterile inoculation loop and inoculated in the YPD liquid medium for activation twice. The yeast cell pellets were collected by centrifugation at 10,000× *g* (Thermo Sorvall LYNX6000, Thermo Fisher Scientific, Waltham, MA, USA) at room temperature (22–25 °C) for 5 min and washed twice with sterile saline solution. The harvested pellets were quickly frozen in liquid nitrogen and then sent to Shanghai Majorbio Bio-pharm Technology Co., Ltd. (Shanghai, China) for DNA extraction and whole-genome sequencing. The DNA meeting quality standards was paired-end sequencing on the second-generation high-throughput sequencing platform Illumina HiSeq×10 (Illumina, Inc., San Diego, CA, USA). After removing low-quality data, the software SOAPDenovo 2 was used to perform whole-genome assembly on the clean data to obtain the optimal assembly result. These assembled genes were functionally annotated using GO, COG, KEGG, and other databases.

### 2.2. Preparation of Fermentation Medium

The fresh white radish was purchased from a local market in Kunming, China. The initial step involved homogenizing the vegetables using a juicer to prepare the vegetable juice medium. The resulting radish mixture was then filtered using a gauze (JYZ-E3C, Joyoung Co., Ltd., Jinan City, Shandong Province, China) and transferred into individual PE bottles, ensuring no headspace remained. To achieve sterilization through filtration, the extracted radish juices underwent high-pressure processing (HPP) (Xiecheng, XC-LF3AH, Shanghai, China) at 600 MPa for a duration of 5 min, while maintaining a temperature of approximately 4 °C [22]. The HPP-treated radish juices were subsequently centrifuged at 10,000× *g* (Thermo Sorvall LYNX6000, Thermo Fisher Scientific, Waltham, MA, USA) at room temperature (22–25 °C) for 20 min, followed by vacuum filtration through a 0.22 μm polyester membrane (BS-500-XT, Biosharp Life Sciences Co., Ltd., Lu’an City, Anhui Province, China) for sterilization purposes. The final vegetable juice was stored at 4 °C for later use. Approximately 300 mL of pale yellow, transparent juice was obtained from 1 kg of radish. The blank control medium was a yeast extract peptone dextrose (YPD) medium, which consisted of 20 g/L glucose, 20 g/L peptone, and 10 g/L yeast extract. Glucose, peptone, and yeast extract (Shanghai Yuanye Bio-Technology Co., Ltd., Shanghai, China) were used as standards for identification and quantification.

### 2.3. Determination of Physical and Chemical Properties of Fermented Radish Juice

#### 2.3.1. Yeast Growth Performance Monitoring

The yeast cells were pre-cultured on YPD agar plates by spreading. Single colonies from the plate were selected and inoculated at a concentration of 1.0 g/L into YPD medium and cultured at 30 °C with shaking at 210 rpm for 72 h for pre-cultivation. Subsequently, the yeast cells were washed twice with sterile water, and the resulting material was collected and resuspended in radish juice. The initial optical density (OD_600_) of the inoculum was adjusted to 0.4. The culture plate was incubated at 30 °C with a shaking speed of 250 rpm for a period of 96 h using a microbiological reader (LP600-SN, Agilent Technologies, Santa Clara, CA, USA), with optical density measurements at 600 nm (OD_600_) being recorded at 20 min intervals. A blank culture medium was used as a control under identical conditions, showing no significant difference in OD values.

#### 2.3.2. pH, Total Soluble Solid, and Sugar Determination

To determine the pH value of the fermented vegetable juice, we used a pH meter (Mettler Toledo, Shanghai, China). Total soluble solids (TSS) were measured at 24–26 °C using a Brix refractometer and the results are expressed in °Brix. Sugars were extracted from vegetable juices by employing a method described in previous research [23]. Initially, 1.0 mL of vegetable juice was mixed with 10 mL of milli-Q water, followed by the addition of 500 µL of Carrez I (15% *w*/*v* K_4_[Fe(CN)_6_]) and 500 µL of Carrez II (30% *w*/*v* ZnSO_4_) using a vortex mixer. After 30 min of incubation, it was centrifuged at 22,000× *g* for 20 min at a temperature of 4 °C. The supernatant was collected and filtered through a 0.45 µm syringe filter. The prepared samples were analyzed using an HPLC system (Agilent 1260, Agilent Technologies, Santa Clara, CA, USA), which was equipped with an ELSD detector (1260 Series, Agilent Technologies, Santa Clara, CA, USA) and featured a C18 column (Asahipak NH2P-50 4E, Shodex, Tokyo, Japan) in conjunction with a Prevail C18 guard cartridge. The analysis was performed at a flow rate of 1.0 mL/min and an injection volume of 5.0 µL, using isocratic elution with a water–acetonitrile mixture (25:75, *v*/*v*). Glucose, fructose, and sucrose (Shanghai Yuanye Bio-Technology Co., Ltd., Shanghai, China) were used as standards for identification and quantification.

#### 2.3.3. Soluble Protein

The soluble protein content of the fermentation broth was determined using a BCA protein assay kit purchased from Beyotime (Shanghai, China). The protein content determination was conducted according to the instructions. The absorbance of the sample was measured using a microplate reader spectrophotometer (UV 5100B, Shanghai, China) at 562 nm.

### 2.4. Determination of Glycosidase Activity

To characterize the glycosidase activity, the strain was inoculated in the YPD liquid medium containing 5.0 g/L different glycoside instead of glucose at an inoculum concentration of 1.0 g/L. These cultures were incubated at 30 °C for 72 h. Supernatants were collected by centrifugation at 4 °C, 10,000× *g* (Thermo Sorvall LYNX6000, Thermo Fisher Scientific, Waltham, MA, USA), 10 min, to determine the enzyme activity. The identified enzyme activities were further validated in fermented radish juice. Glucosidase activity in fermentation broth was assayed based on the spectrophotometric method. A total of 200 μL fermentation broth was mixed with 750 μL citrate phosphate (0.1 mol/L citric acid and 0.2 mol/L dipotassium hydrogen phosphate, pH 5.0) buffer solution. β-glucosidase, α-mannosidase, and α-galactosidase activities were assayed by adding 250 μL 1 mmol/L of *p*-nitrophenyl-β-glucopyranoside, *p*-nitrophenyl-α-D-mannopyranoside, and *p*-nitrophenyl-α-d-galactopyranoside in the mixture solution, respectively [24,25,26]. After incubation at 37 °C for 30 min, reactions were terminated with 1 mL 1.0 M sodium carbonate. The enzyme activity was determined using the standard curve of *p*-nitrophenyl, which is the product of enzymatic hydrolysis and has a maximum optical absorption at 400 nm. The enzyme unit was defined as the amount of enzyme that, under the aforementioned conditions, had hydrolyzed 1 mol of substrate (*p*-nitrophenyl-β-glucopyranoside, *p*-nitrophenyl-α-d-mannopyranoside, and *p*-nitrophenyl-α-d-galactopyranoside) per second to release 1 mol of p-nitrophenol, expressed as 1 nkat/mL.

### 2.5. Proteome Analysis

For proteome analysis, the sample preparation method was consistent with that of transcriptome sequencing, and the services of Biomarker Technologies Co., Ltd. (Beijing, China) were engaged to employ TMT™ (Tandem Mass Tag™) technology for a comprehensive workflow that included RNA extraction, quality shearing, statistical quality control, library construction, and subsequent sequencing. The resulting protein data were subjected to comparative analysis against the Swiss-Prot database to identify differentially expressed proteins across various groups. Quantitative information pertaining to these differential proteins was normalized to ensure the reliability of subsequent analyses. Proteins exhibiting significant differential expression were identified based on the stringent criteria of |*log*_2_*FC*| ≥ 1 and a *p* ≤ 0.05. To functionally annotate the identified differentially expressed proteins (DEPs), databases such as GO and KEGG were utilized. These annotations facilitated a differential enrichment analysis, providing insights into the biological processes, molecular functions, and cellular components associated with the DEPs, as well as their involvement in specific pathways.

### 2.6. Transcriptome Analysis

To identify the genes responsible for coding the glucosidase, the strain was inoculated in the radish juice medium at an inoculum volume of 1.0 g/L and incubated at 30 °C, 200 rpm. The logarithmic growth-phase samples were collected and frozen in liquid nitrogen. The extraction, quality shearing, statistical quality control, library construction, and sequencing of RNA were completed by Biomarker Technologies Co., Ltd. The quality-controlled clean reads were mapped to the reference genome of *Rhodotorula mucilaginosa* (GenBank: GCA_032719295.1) to ensure alignment accuracy. Subsequently, the assembly of these aligned reads was conducted using StringTie (version 1.38.2, Baltimore, MD, USA) [27], a robust tool for transcriptome assembly. To normalize the transcriptome data, both the number of assembled reads and the transcript length were adjusted to account for differences in sequencing depth and gene length. The normalization process employed the FPKM method [28], which leverages the maximum flow algorithm within StringTie to provide a measure of gene expression that is independent of transcript length and sequencing depth. To identify differentially expressed genes (DEGs), the DESeq software package(version 2.2.3, Heidelberg, Germany) was utilized, applying the stringent criteria of |*log*_2_*FC*| ≥ 1 and a *p* ≤ 0.05 to filter significant expression changes. Furthermore, to elucidate the biological significance of these DEGs, their involvement in various signaling pathways was investigated using databases such as KEGG, offering insights into the molecular mechanisms underlying the observed gene expression patterns.

### 2.7. Effects of Commercial Glycosidase and Yeast Fermentation on the Flavor Characteristics of Radish Juice

In order to identify the specific aromas released by glycosidase derived from the yeast in the fermentation of radish juice, we added 33.3 × 10^−2^ nkat/mL [29] commercial β-glucosidase (Shanghai Yuanye Bio-Technology Co., Ltd., Shanghai, China. CAS 9001-22-3), α-mannosidase (Shenzhen Simiga Biotechnology Co., Ltd., Shenzhen, Guangdong Province, China. CAS 9025-42-7), and α-galactosidase (Shanghai Eon Chemical Technology Co., Ltd., Shanghai, China. CAS 9025-35-8) to treat fresh radish juice. The produced volatiles in the treated and yeast fermented radish juices were identified and compared by GC-MS. The identified substances produced by enzymatic hydrolysis were accurately quantified by the external standard curve and quantified by comparison of their chromatographic peaks areas with that corresponding to the external standard, and the results are expressed as μL/L.

The extraction, identification, and quantification of volatiles were performed using the methods described in a previous study [30]. Volatile compounds were extracted from radish juices, yeast-fermented radish juice and commercial glycosidase-treated radish juice, using headspace solid-phase microextraction. We mixed cell-free broth (3 mL) with NaCl (1 g) by vortexing. Then, we added 30 μL of 50 μL/L 4-methyl-1-pentanol as an internal standard. After equilibration at 40 °C and 500 rpm for 15 min, 50/30 µm (Zhenzheng Analytical Instrument Co., Ltd., Qingdao, Shandong Province, China), SPME fibers were used to extract volatile compounds under the same conditions for 40 min. Then, we inserted the extraction needle into the GC inlet and performed thermal desorption at 250 °C for 5 min. A GC-MS system equipped with a DB-5 MS column (30 m × 0.32 mm × 0.25 μm, Agilent Technologies, Inc., Santa Clara, CA, USA) was used to separate and detect volatiles. The temperature program of the GC column oven was as follows: First, the temperature rose to 40 °C and was held for 3 min, then increased at a rate of 4 °C/min to reach 120 °C, where it was held for another 3 min. Secondly, the temperature rose at a rate of 10 °C/min to 230 °C and was held there for 3 min; finally, it cooled back down to 40 °C. High-purity helium (99.99%) was used as the carrier gas with a flow rate of 3.0 mL/min. The mass spectrometry conditions were set with an electron energy of 70 eV and a scanning range of 35–500 m/z. The MS ion source temperature was maintained at 230 °C and the quadrupole temperature at 280 °C. Each sample underwent three parallel experiments.

The odor activity values (OAVs) of volatile compounds were calculated to identify key odorants. Compounds with OAV > 1 were considered core aroma-active compounds. OAV was calculated as follows: OAV = concentration/threshold value of the compound in the fermented vegetable juice sample. Threshold values for each compound were obtained from references.

### 2.8. Statistical Analysis

The significance difference of results was analyzed by a one-way analysis of variance (ANOVA) and Tukey’s test (*p* < 0.05) using IBM SPSS (version 20.0, Chicago, IL, USA). Data are shown as the mean value ± standard deviation. Genome-wide data analysis was performed using the Majorbio cloud platform (https://cloud.Majorbio.com/page/tools/) (accessed on 1 August 2023). Gene annotation was performed with reference to the carbohydrate-active enzyme (CAZy) database. The transcriptome and proteome analyses of raw data were performed via the BMKCloud (www.biocloud.net) (accessed on 1 November 2023) online platform.

## 3. Results and Discussion

### 3.1. Identification of Glycosidase-Encoding Genes in Yeast

The genome of *R. mucilaginosa* was sequenced using the Illumina Hiseq platform. The DNA library generated a total of 47.7 Mb of raw data. After removing low-quality reads, 45.8 Mb of clean-read data was obtained for genome assembly (Q20 > 95%, Q30 > 90%). The genome size of this yeast was 20.03 Mb. The final assembly consisted of 442 scaffolds with an N50 length of 174,712 bp and a GC content of 60.58%. A total of 5262 encoding-genes were identified in the genome, with an average length of 2936 bp and an average GC content of 61.43%, accounting for 77.14% of the genome. As shown in Table 1, according to CAZy, a total of 37 glycoside hydrolases (GHs) were annotated in yeast strain I2002. Among these, only nine of their corresponding coding genes could be identified. After blasting in NCBI, these nine genes included β-glucosidase-encoding gene *bglX*, α-glucosidase maltase gene *malL*, mannan oligosaccharide α-1,2-mannosidase, alpha-mannosidase *MNS1_2*, 6-phosphohydrolase gene *sacA*, d-α-d-mannose-1-phosphate guanylyltransferase gene *ManB*, trehalase gene *treA*, α-amylase genes *amyA*, and α-1,2-mannosidase gene *MNS3*. Functional annotations mainly included trehalase, α-mannosidase, β-glucosidase, α-galactosidase, etc., indicating that the I2002 strain could produce related glycosidases. By studying the metabolism of trehalose in transformed *Listeria monocytogenes*, previous researchers found that trehalose undergoes acidification to generate trehalose-6-phosphate, and then trehalose is decomposed into glucose and glucose-6-phosphate by the *treA* gene [31]. They also found that the main activity of *amyA* gene is to catalyze the hydrolysis of internal α-1,4-glycosidic bonds in starch through a double displacement mechanism, mainly producing glucose and maltose [32].

### 3.2. Fermentation Performance of Yeast in Radish Juice

The premise for yeast to metabolize and produce glycosidase was growing it in radish juice. Therefore, the growth profile and fermentation characteristics were initially determined. As shown in Figure 1A, *Rhodotorula mucilaginosa* consumed nutrients and grew well in radish juice. Around 15 h of cultivation, the strain grew rapidly and entered the logarithmic phase of growth, with a maximum growth rate reaching 7.9 × 10^−4^ OD/min. After 36 h, it gradually entered a stable phase. The pH, TSS, soluble protein, and sugar content of the fermentation broth of the strain in radish juice were monitored (Figure 1B–D). It was found that the soluble protein concentration in fermentation broth increased gradually within 72 h, indicating that this strain may produce some extracellular protein or enzymes in the radish juice. The pH in the fermentation broth dropped from 6.7 to 4.0 and then gradually increased to 4.5. This phenomenon may be attributed to the strain metabolizes nutrients to produce some acidity by products in the early fermentation stage, resulting in the pH decrease. At 24–36 h of fermentation, the TSS content dropped rapidly and the pH value showed a slight upward trend. At this time, the yeast entered the logarithmic phase of growth and the number of cells gradually increased. After 72 h of fermentation of radish juice, the TSS content dropped to 0.2%, and the nutrients in the culture medium were insufficient to support the growth of the strain. Therefore, the acidic metabolites accumulated in the early stages were consumed by the yeast as nutrients, resulting in a weak pH rising trend. A similar result was also observed in another study conducted by [33]. Regarding sugars, this yeast primarily consumed glucose and fructose. Sucrose was undetected in this case, as vegetables mainly contain reducing sugars, which could provide ideal carbon sources for microbial growth during fermentation [29]. During the fermentation process, both glucose and fructose contents gradually decreased, but glucose had a faster consuming rate (Figure 1D). All glucose was consumed within 36 h, while there were around 2 g/L fructose unconsumed at the end of fermentation. In the study of the chemical composition of sauerkraut produced from different varieties of cabbage, it was found that the degree of utilization of glucose was higher than that of fructose, which may be related to the fact that microorganisms such as lactic acid bacteria and yeast utilize glucose first [34].

### 3.3. Glycosidase Activity

Based on the sequencing and analysis of the whole genome of *R. mucilaginosa*, it was speculated that this strain had a high probability of containing glycosidase-related genes; so, the PNPG colorimetric method was used to verify whether the strain had glycosidase activity. In the YPD liquid medium inoculated with *R. mucilaginosa*, no glycosidase activity was detected. After supplemented with glycoside, three kinds of glycosidase activity were detected, including β-glucosidase, α-mannosidase, and α-galactosidase (Figure 2A). Moreover, the enzymatic activity measured at 72 h was significantly higher than that tested at 24 h. Studies had found that the best substrate for β-D-glucosidase production by *Trichoderma reesei* S12 was pNPG, possibly because the substrate induced glycosidase production [35]. This result implies that the presence of glycoside in the medium was the trigger of this strain to produce relative glycosidases. Among these glycosidases, β-glucosidase has the highest activity, reaching 28.0 × 10^−2^ nkat/mL at 72 h, followed by α-mannosidase at 12.2 × 10^−2^ nkat/mL, and α-mannosidase with the lowest activity at 4.30 × 10^−2^ nkat/mL, which is also consistent with the results in the literature that *R. mucilaginosa* is a strain with high β-glucosidase production [36].

To further identify the glycosidase produced by yeast in radish juice, the activity of these three glycosidases was assayed (Figure 2B). The activity of α-galactose maintained a low level (1.33 × 10^−2^ nkat/mL) from 24 to 72 h. The activities of β-glucosidase and α-mannosidase were significantly reduced. Specifically, the activity of α-mannosidase decreased from 44.3 × 10^−2^ nkat/mL at 24 h to 17.1 × 10^−2^ nkat/mL at 72 h. α-galactosidase primarily acts on α-1,6-galactoside linkages [37] and α-mannosidase participated in eukaryotic protein N-glycan modification, as well as hydrolysis of mannose α-1,2, α-1,3, α-1,6 glycosidic bonds [38,39]. Moreover, the glycosidase activity of strains largely depended on the presence of relative glycose in the fermentation medium. Therefore, the radish juice may contain more mannoside-binding compounds. Additionally, the decrease in glycosidase activity from 24 to 72 h may be caused by that fact that the enzyme-binding sites were occupied by glycoses in radish juice during the fermentation process.

### 3.4. Proteome Analysis

To further identify the potential intracellular glycosidase produced by this yeast, the intracellular proteome was also analyzed. A total of 60 significantly different proteins were detected from samples of two different stages of fermented radish juice by *Rhodotorula mucilaginosa* I2002. Among them, 24 differential proteins (40%) were significantly up-regulated and 36 differential proteins (60%) were significantly down-regulated. GO functional annotation was performed on DEPs. The results are shown in Figure 3A. The cellular component mainly included six membrane proteins and six membrane part proteins; the molecular function mainly encompassed four catalytic activity proteins and two binding proteins; biological process primarily included two single-organism process proteins and two localization proteins. The GO enrichment analysis was performed on differential proteins. In the two groups of samples, histone acetyltransferase SAGA complex, integral components of membranes, Ada2/Gcn5/Ada3 transcription activator complex, Smc5-Smc6 complex, H4 histone acetyltransferase complex, and other components underwent significant changes.

Through the KEGG functional annotation of all differential proteins (Figure 3B), it was found that differentially expressed proteins were mainly enriched in pathways such as metabolism and genetic information processing. Among them, four differential proteins were enriched in oxidative phosphorylation, three differential proteins were enriched in the biosynthesis of amino acids, three differential proteins were enriched in ribosome, two differential proteins were enriched in protein processing in endoplasmic reticulum, two differential proteins were enriched in cysteine and methionine metabolism, and two differential proteins were enriched in fatty acid metabolism, fructose, and mannose metabolism, and carbon metabolism each enriched one differential protein. Finally, two important differentially expressed proteins related to sugar metabolism were detected. One was mannose-6-phosphate isomerase (*manA*, EC: 5.3.1.8), which has the ability to convert mannose-6-phosphate into fructose-6-phosphate and is an intermediate in the glycolysis pathway [40]. So, mannose-6-phosphate isomerase plays an important role in sugar metabolism. Another one was sucrose-6-phosphate hydrolase (*sacA*, EC: 3.2.1.26), which acts on the glycosidic bond at the non-reducing end to release fructose [41]. Many glycosidases (EC: 3.2.1) were annotated in the starch and sucrose metabolism signaling pathways of the differential protein KEGG, among which fructosidase (EC: 3.2.1.26) was significantly upregulated. These findings suggest that these are related to starch. Differential genes related to sucrose and mannose metabolism may played a crucial role in regulating the accumulation of glycoside flavor substances.

### 3.5. Transcriptome Analysis

To further verify the correlation between glycoside metabolism and flavor formation during fermentation, the transcriptome of yeast was also analyzed. The DESeq2 [42] differential analysis software was used to standardize the gene read count values in the samples, screen differentially expressed genes, and perform a differential analysis on differential groups with biological repeats; for differential groups without biological repeats, the edgeR [43] software was used to perform for differential analysis. During the detection of differentially expressed genes, the fold change represents the ratio of expression levels between two sample groups. The false discovery rate (FDR) was obtained by correcting the *p*-value of the difference significance. Fold change ≥ 1 and FDR < 0.01 were used as the screening criteria for differentially expressed genes, which indicated the significance of the difference. For the convenience of comparison, the logarithmic value was taken as the difference multiple and expressed as *log_2_FC*. The larger the absolute value of *log_2_FC*, the smaller the FDR value, and the more obvious the difference between the two groups of samples. If the differential gene *log_2_FC* > 0, it was considered to be up-regulated; on the contrary, if the differential gene *log*_2_*FC* < 0, the differentially expressed gene was down-regulated. A total of 594 significantly different genes were detected in two different fermentation time groups of *Rhodotorula mucilaginosa* I2002, including 359 significantly up-regulated genes and 270 significantly down-regulated genes.

Through KEGG database annotation, a total of 52 pathways were annotated in *Rhodotorula mucilaginosa* I2002, and the 20 most significantly enriched pathways were selected to develop the KEGG enrichment bubble map (Figure 4B). Among them, the pathways with the largest number of enriched differential genes included carbohydrate metabolism enriched with 14 differential genes, biosynthesis of amino acids enriched with 10 differential genes, purine metabolism peroxisome enriched with 10 differential genes, cell cycle-yeast enriched with 8 differential genes, 2-Oxocarboxylic acid metabolism enriched with 7 differential genes, pyruvate metabolism enriched with 7 differential genes, and citrate cycle enriched with 7 differential genes.

Differential expressed genes related to glycosidase metabolism were further analyzed, including starch and sucrose metabolism pathways and glycolysis pathways. These differentially expressed genes are important regulators involved in the flavor of fermented foods. Table 2 shows the differential gene expression of glycosidase in the two groups of logarithmic and plateau phase samples. As the fermentation time increased, the expression of the three glycosidase genes annotated by the KEGG pathway significantly increased (*bglX* and *malL*). It could be inferred that glycosidase gene expression was activated during the fermentation stage of radish juice, and as the fermentation time prolongs, the glycosidase gene expression increased. These glycosidase genes participated in metabolic hydrolysis to release aroma precursors in fermented vegetables. For example, glucosidase was the most important hydrolase involved in the hydrolysis of terpenes and phenolic glycosides, and was responsible for free terpenes, flavonoid aglycones, and quinoa [44]. Previous research has established that the fundamental mechanism of hydrolysis of α-1,6-glucosidic bonds by α-glucosidase maltase gene (*malL*) was essentially the same as that of other starch-degrading enzymes belonging to family 13 (α-amylase family), proceeding via hydrolysis with the cleavage of the glycosidic bond [45]. Studies have shown that the β-glucosidase gene (*bglX*) hydrolyzes β-glycosidic linkages in nitrophenyl-β-glycosides and exhibits a greater activity on galactose-containing substrates than glucose derivatives, but does not hydrolyze cellobiose, maltose, or laminarin [46].

### 3.6. Identification of Aroma Compounds Released by Glycosidase Produced by Yeast

Radish juice was subjected to fermentation using yeast and treated with three distinct commercial glycosidases. GC-MS technology was employed to investigate the changes in the volatile aroma profile of the radish juice (Figure 5). The detected volatile compounds are specific chemical substances in radish juice, which form the basis of the aroma component in the flavor profile of radish juice. A total of 12 flavor substances were detected in fresh radish juice, and 17 flavor substances were detected in fermented radish juice. A total of 26, 24, and 28 species were detected in the three enzyme treatment groups of α-galactosidase, β-glucosidase, and α-mannosidase, respectively. A total of 11 identical substances were detected in fermented radish juice and three types of enzyme-treated radish juice. Except for the vanilla-flavored 1,2-dimethoxybenzene, which was not detected in the α-mannosidase-treated group, all other groups contained this substance. Among them, the main flavor of geranylacetone, methyl butyrate, methyl 2-methylbutanoate, 4-methyl-1-pentanol, and other substances was fruity. 1-Dodecanol, nonanal, and 1-nonanol had a soapy smell. 2-ethyl-1-hexanol and 4-methyl-1-pentanol were ethereal and fruity. After comparing the red yeast fermentation results with the fresh radish juice (Table 3), it can be concluded that the flavor substances produced by the strain and by α-galactosidase, β-glucosidase, and α-mannosidase included cis-2-pentenol (green scent), hyacinthin (floral, honey), geranylacetone (fruity), and 1-dodecanol (soapy, sweet). The reason for the lower variety and quantity of flavor compounds detected in radish juice fermented with Rhodotorula mucilaginosa compared to those treated with commercial glycosidases may be attributed to the relatively fixed metabolic pathways, lower enzyme activity, and specificity of R. mucilaginosa, as well as the influence of fermentation conditions. Subsequently, optimizing the fermentation conditions could enhance the diversity of flavors. The OAVs of the substances detected in the samples were mostly greater than 1.0, indicating that the added glycosidase had a greater contribution to the flavor of vegetables.

In the next step, a GC-MS apparatus was utilized in conjunction with four flavor compound standards (concentrations ranging from 10 to 200 μL/L) to establish a calibration curve. This curve was employed to perform the quantitative analysis of the target compounds within the sample via the external standard method. The structural formulas and retention times of these four flavor compounds are depicted in Figure 5. The retention times of the flavor compounds in the samples were found to be congruent with those of the standards, suggesting that several substances were enzymatically released by the glycosidase treatment. The aforementioned calibration curve was employed to analyze the samples and quantified the concentration of flavor compounds in the product, as detailed in Table 3. Among the three commercial samples treated with glycosidases, the content of cis-2-pentenol was markedly higher compared to the other three flavor compounds, peaking at 231 μL/L. After a 72 h fermentation period, only the content in the MAN (α-mannosidase) treatment group exhibited a slight increase. Within the MAN treatment group, the levels of hyacinthin and cis-2-pentenol were significantly elevated compared to the other two groups, with a maximum concentration of 234 μL/L being observed. Following 72 h of fermentation, the content of 1-dodecanol increased notably. This outcome aligned with the observed upregulation of glycosidase gene expression (*bglX* and *malL*), indicating that these genes were capable of hydrolyzing and liberating endogenous aromatic compounds from vegetable substrates.

## 4. Conclusions

The present study offers a novel exploration into the impact of glycosidases produced by the red yeast *Rhodotorula mucilaginosa* on the release of bound flavor compounds in fermented white radish. Through whole-genome sequencing, a total of 37 glycosidase genes were identified within *R. mucilaginosa*, with 9 coding genes being annotated. Utilizing the PNPG colorimetric method, the presence of α-galactosidase, β-glucosidase, and α-mannosidase activities in the yeast strain was confirmed. A significant finding of this study is the up-regulation of the β-glucosidase gene (*bglX*) and the α-glucosidase maltase gene (*malL*) following the inoculation of *R. mucilaginosa* in fermented radish juice. This up-regulation was accompanied by an increase in the content of specific flavor compounds, indicating that these glycosidases are actively involved in the hydrolysis and release of bound flavor compounds in vegetables.

Furthermore, four flavor substances—cis-2-pentenol, hyacinthin, geranylacetone, and 1-dodecanol—were identified in radish juice that had been treated with both red yeast and commercial glycosidase. These glycosides were isolated at various stages of the fermentation process. An analysis of both the expression levels of the glycosidase genes and the concentrations of the flavor substances revealed a positive correlation, wherein the content of the flavor substances rose in tandem with the increased expression of the glycosidase genes.

The novelty of this study lies in the use of multi-omics approaches to comprehensively analyze the role of glycosidases produced by *R. mucilaginosa* in the release of flavor compounds during vegetable fermentation. This comprehensive analysis provides new insights into the formation mechanism of flavor compounds in fermented vegetables and offers a theoretical basis for improving the quality of fermented vegetable products through the targeted manipulation of glycosidase activities.

## Figures and Tables

**Figure 1 foods-14-01263-f001:**
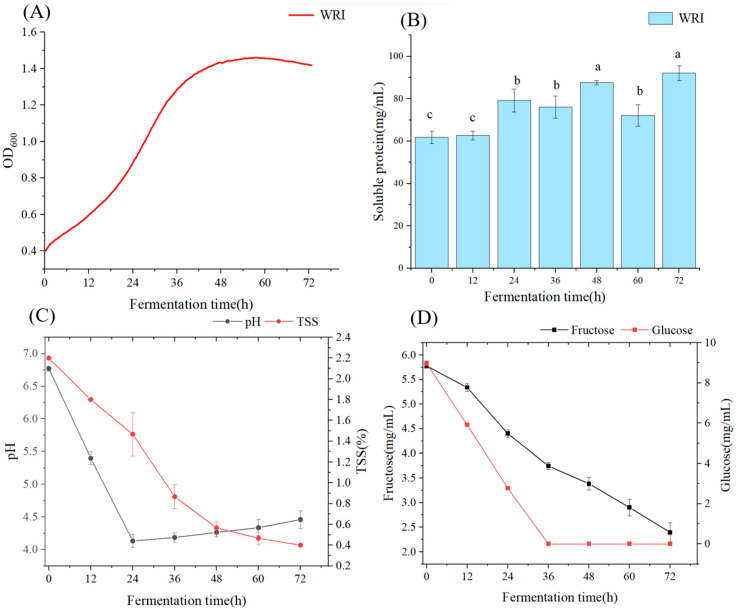
Physicochemical properties of the strain in radish juice fermentation: (**A**): growth curve; (**B**): soluble protein content; (**C**): pH values and total soluble solid; and (**D**): fructose and glucose content. WRI: yeast fermented in white radish juice. Different lowercase letters indicate significant differences in flavor substances among different samples (*p* < 0.05).

**Figure 2 foods-14-01263-f002:**
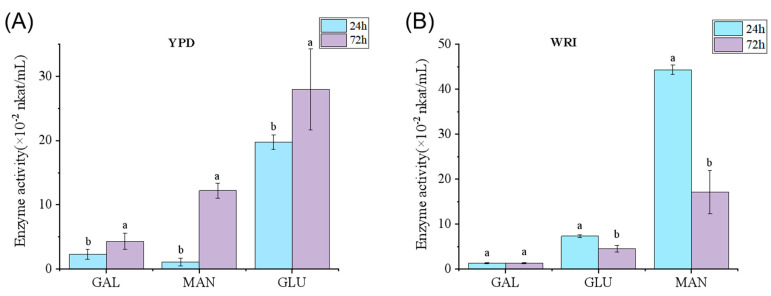
Determination of glycosidase activity. (**A**): Glycosidase activities in YPD medium. (**B**): Glycosidase activities in fermented white radish juice. MAN: α-mannosidase; GLU: β-glucosidase; GAL: α-galactosidase. Different lowercase letters indicate significant differences in flavor substances among different samples *(p* < 0.05).

**Figure 3 foods-14-01263-f003:**
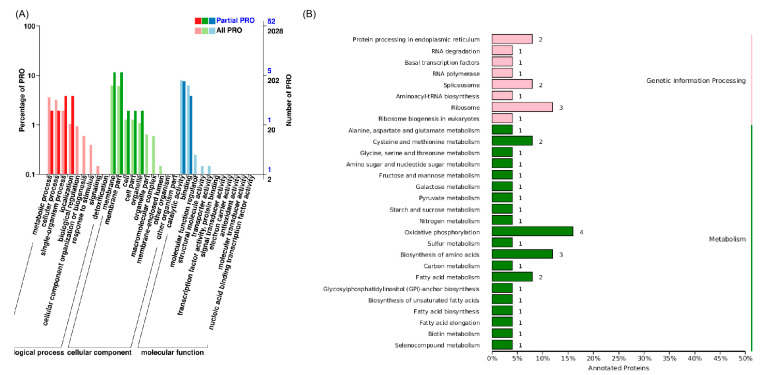
Differential proteins. (**A**) GO database annotation and enrichment. (**B**) KEGG pathway classification bar chart.

**Figure 4 foods-14-01263-f004:**
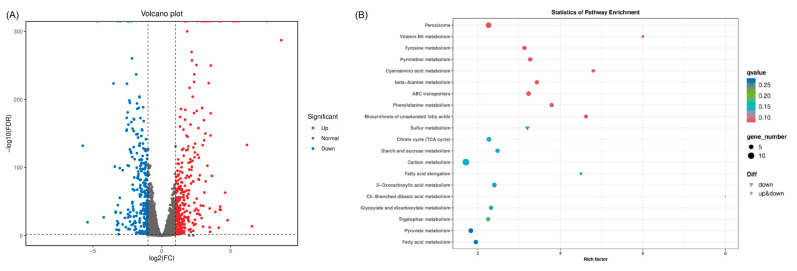
Differentially expressed genes. (**A**): Volcano map; (**B**): KEGG pathway enrichment bubble map.

**Figure 5 foods-14-01263-f005:**
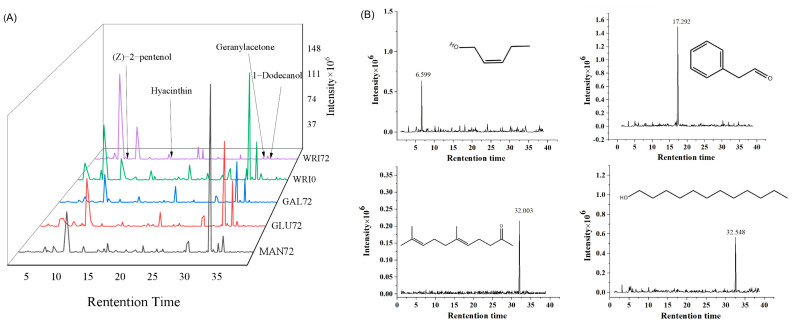
Volatile flavor compounds in fermentation broth. (**A**) Total ion flow diagram. (**B**) The structural formula and retention time of four common flavor compounds. WRI0: Blank white radish juice; WRI72: yeast fermented in radish juice for 72 h; MAN72: α-mannosidase treatment of radish juice for 72 h; GLU72: β-glucosidase treatment of radish juice for 72 h; GAL72: α-galactosidase treatment of radish juice for 72 h.

**Table 1 foods-14-01263-t001:** Glycoside-encoding gene annotation statistics.

NR Hit	Gene Name	Family	Family Description
KWU45371.1	*bglX*	GH3	beta-glucosidase
TKA50830.1	*MNS1_2*	GH47	alpha-mannosidase
TKA53735.1	*sacA*	GH32	fructan beta-(2,1)-fructosidase
KWU44931.1	*manB*	GH2	beta-galactosidase; beta-mannosidase
TKA58479.1	*treA*	GH37	alpha, alpha-trehalase
KWU41236.1	*malL*	GH13_40	GH13’s subfamily
TKA52136.1	*amyA*	GH13_5	GH13’s subfamily
TKA57373.1	*MNS3*	GH47	alpha-mannosidase
TKA50408.1	*SAE1*	GH5_8	GH5’s subfamily

**Table 2 foods-14-01263-t002:** Differential gene expression of glycosidase.

Gene ID	Gene Name	L24-Count	L72-Count	L24-FPKM	L72-FPKM	log2FC	Regulated	KEGG_ Annotation
Gene0092	*bglX*	482	1443	4.79	12.39	1.11	up	K05349 beta-glucosidase [EC:3.2.1.21]
Gene0077	*bglX*	291	816	2.95	7.17	1.17	up	K05349 beta-glucosidase [EC:3.2.1.21]
Gene0853	*malL*	343	1791	4.55	20.52	2.26	up	K01182 oligo-1,6-glucosidase [EC:3.2.1.10]

**Table 3 foods-14-01263-t003:** Flavor substances produced by commercial glycosidase and I2002 processing in radish juice.

Code	CAS	Compounds	Concentrations (uL/L)									Description
			WRI 0	WRI 24	WRI 72	GAL 24	GAL 72	GLU 24	GLU 72	MAN 24	MAN 72	
1	1576-95-0	(Z)-2-pentenol	-	34.88 ±3.71 ^b^	-	231.23 ±16.38 ^a^	197.14 ±17.99 ^a^	231.16 ±12.64 ^a^	196.08 ±33.79 ^a^	181.34 ±9.98 ^a^	202.56 ±6.69 ^a^	green scent
2	122-78-1	Hyacinthin	-	38.17 ±4.14 ^b^	30.08 ±10.42 ^b^	-	33.55 ±0.52 ^b^	-	27.68 ±2.72 ^b^	228.24 ±1.27 ^a^	234.54 ±5.42 ^a^	floral, honey
3	689-67-8	Geranylacetone	-	13.27 ±0.64 ^a^	12.45 ±0.22 ^a^	21.32 ±0.85 ^a^	22.66 ±0.4 ^a^	17.34 ±1.51 ^a^	19.63 ±2.05 ^a^	5.93 ±0.72 ^a^	4.72 ±0.46 ^a^	fruity
4	112-53-8	1-Dodecanol	-	6.76 ±2.03 ^c^	9.40 ±0.94 ^b^	19.08 ±1.83 ^b^	19.00 ±3.48 ^b^	16.67 ±4.82 ^b^	21.64 ±3.98 ^b^	5.94 ±1.21 ^c^	60.51 ±6.88 ^a^	soapy, sweet
5	623-42-7	methyl butanoate	+	+	-	+	+	+	+	+	+	fruity
6	868-57-5	methyl 2-methylbutanoate	+	+	+	+	+	+	+	+	+	fruity
7	626-89-1	4-methyl-1-pentanol	+	+	+	+	+	+	+	+	+	fruity, ethereal
8	3658-80-8	dimethyl trisulfide	+	+	-	+	+	+	+	+	+	sulfuric, cabbage-like
9	104-76-7	2-ethyl-1-hexanol	+	+	+	+	+	+	+	+	+	fruity, ethereal
10	124-19-6	nonanal	+	+	+	+	+	+	+	+	+	citrus-like, soapy
11	91-16-7	1,2-Dimethoxybenzene	+	+	+	+	+	+	+	-	-	vanilla
12	143-08-8	1-nonanol	+	+	-	+	+	+	+	+	+	soapy, fruity

Notes: Symbols mean negative (-) or positive (+). WRI 0: Blank white radish juice; WRI 24: yeast fermented in radish juice for 24 h; WRI 72: yeast fermented in radish juice for 72 h; MAN 24: α-mannosidase treatment of radish juice for 24 h; MAN 72: α-mannosidase treatment of radish juice for 72 h; GLU 24: β-glucosidase treatment of radish juice for 24 h; GLU 72: β-glucosidase treatment of radish juice for 72 h; GAL 24: α-galactosidase treatment of radish juice for 24 h; GAL 72: α-galactosidase treatment of radish juice for 72 h. Different lowercase letters indicate significant differences in flavor substances among different samples (*p* < 0.05).

## Data Availability

The original contributions presented in this study are included in the article. Further inquiries can be directed to the corresponding author.

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
