# Peer review of "Effect of Glycosidase Production by Rhodotorula mucilaginosa on the Release of Flavor Compounds in Fermented White Radish"

_foods, 2025, doi:10.3390/foods14071263_

Round 1

Reviewer 1 Report

Comments and Suggestions for Authors

Dear Authors,

I have carefully reviewed your manuscript and would like to acknowledge the robust experimental design and rigorous methodological approach. The study is well-structured, and the conclusions are well-aligned with the results. While the topic may not present groundbreaking novelty, the work falls within the scope of Foods and contributes valuable insights into the enzymatic activity of Rhodotorula mucilaginosa in fermented products. Additionally, the use of Rhodotorula as a potential platform is interesting, given its ability to produce pigments, which could have further implications for food applications.

One aspect that could enhance the impact of your study is the inclusion of a control fermentation using only Saccharomyces. Given that Rhodotorula mucilaginosa follows an oxidative metabolism, a direct comparison with a well-known fermentative yeast such as Saccharomyces would provide a more comprehensive understanding of how R. mucilaginosa affects flavor compound release differently from a traditional fermentative pathway. This would allow a clearer interpretation of whether the observed effects are due to enzymatic activity alone or also influenced by metabolic differences.

If a new experimental setup is not feasible, discussing the potential differences between fermentative (Saccharomyces) and oxidative (Rhodotorula) metabolisms and how they might impact glycosidase activity and flavor compound release would help strengthen the discussion.

Minor Comments:

Consider briefly mentioning the pigment production potential of R. mucilaginosa and whether it played any role in the study or could have implications for future research.

Ensure that all comparisons between Rhodotorula and other yeasts are made cautiously, considering their distinct metabolic profiles.

Overall, this manuscript is well-prepared and within the journal's scope. Addressing the suggested points would further strengthen its contribution to the field.

Best regards,

RC

Author Response

Comments 1: A control fermentation using only Saccharomyces. If a new experimental setup is not feasible, discussing the potential differences between fermentative (Saccharomyces) and oxidative (Rhodotorula) metabolisms and how they might impact glycosidase activity and flavor compound release would help strengthen the discussion.

 Response 1: Thank you for this insightful suggestion. We fully agree that a direct comparison between Saccharomyces (fermentative) and Rhodotorula mucilaginosa (oxidative) metabolisms would provide valuable mechanistic insights. However, due to limitations in time and resources required for establishing new axenic fermentation systems under controlled conditions, we regret that additional experimental comparisons cannot be conducted at this stage. Some concerns on the potential differences between fermentative (Saccharomyces) and oxidative (Rhodotorula) have been added in the new revision of this manuscript:

Saccharomyces cerevisiae primarily relies on fermentation, resulting in the production of ethanol, organic acids, and carbon dioxide (Wu, et al., 2021). In environments with low pH and high ethanol concentrations, glycosidase activity is suppressed, making it difficult to hydrolyze bound aroma precursors. Consequently, the main flavor contributions come from fermentation byproducts such as esters and alcohols (Waymark, et al., 2021). In contrast, Rhodotorula spp. utilize oxidative metabolism, which predominantly produces carbon dioxide, water, and ATP. [Line 65-71]

Comments 2: Consider briefly mentioning the pigment production potential of R. mucilaginosa and whether it played any role in the study or could have implications for future research.

 Response 2: Thank you for your insightful suggestion. While R. mucilaginosa is known for carotenoid biosynthesis, its pigment production was not experimentally addressed here, as our investigation centered on oxidative metabolism and enzymatic flavor release mechanisms.

Comments 3: Ensure that all comparisons between Rhodotorula and other yeasts are made cautiously, considering their distinct metabolic profiles. 

Response 3: We appreciate your attention to detail and acknowledge that the article discusses two yeast species: Rhodotorula mucilaginosa and Saccharomyces cerevisiae. As you have noted in Comment 1, we have provided supplementary information on the distinct metabolic profiles of these yeast species.

References:

  1. Wang, X.-C., Li, A.-H., Dizy, M., Ullah, N., Sun, W.-X., & Tao, Y.-S. (2017). Evaluation of aroma enhancement for "Ecolly" dry white wines by mixed inoculation of selected Rhodotorula mucilaginosaand Saccharomyces cerevisiae. Food Chemistry, 228, 550-559. https://doi.org/10.1016/j.foodchem.2017.01.113

Reviewer 2 Report

Comments and Suggestions for Authors

The article investigates how the glucosidase enzyme produced by Rhodotorula mucilaginosa influences the release of flavor compounds in fermented white radish. Using a multi-omics approach, they analyze gene expression and enzyme activity to determine how these enzymes break down bound compounds and enhance the flavor profile of the fermented product.

This is a novel study, as it explores the specific role of Rhodotorula mucilaginosa in the release of flavor compounds in fermented vegetables through the action of its glucosidases. However, some changes in syntax are suggested:

Modify title

“Effect of Glycosidase Produced by Rhodotorula mucilaginosa on the Release of Flavor Compounds in Fermented White Radish.” By “Effect of Glycosidase Production by Rhodotorula mucilaginosa on the Release of Flavor Compounds in Fermented White Radish.”

Modify in abstract

  • "Fermented vegetables are widely favored by consumers due to their unique flavors and rich nutrient profiles. By “Fermented vegetables are highly valued by consumers for their distinct flavors and rich nutritional content.”
  • "However, there are limited reports on the influence of red yeast on flavor development in fermented vegetables." By “However, studies on the impact of red yeast on flavor development in fermented vegetables remain scarce.”
  • "When vegetable juice was treated with commercial glycosidase, elevated levels of cis-2-pentenol, hyacinthin, geranylacetone, and 1-dodecanol were detected, mirroring the results obtained from yeast-fermented vegetable juice." By “The application of commercial glycosidase to vegetable juice resulted in increased levels of cis-2-pentenol, hyacinthin, geranylacetone, and 1-dodecanol, consistent with findings from yeast-fermented vegetable juice.”

Modify in introduction

  • "Fermented vegetables are traditional foods made to extend the shelf life of vegetables and are widely favored by consumers for their unique flavors." By “Fermented vegetables are traditional foods that extend the shelf life of vegetables and are appreciated for their unique flavors.”

Materials and Methods

  • Review the document and abbreviate minutes as min
  • In page 4 line 154 separate 72h as 72 h
  • Check throughout the document that the degrees Celsius are separated from the number (e.g. 32 °C)
  • Check throughout the document that the % symbol is separated from the number (e.g. 32 %)

  • In the preparation of the fermentation medium, the authors do not clearly specify the controls used. It is recommended that they be added.
  • Why was 600 MPa chosen for high pressure processing (HPP)?
  • It is claimed that yeast significantly increased the levels of β-glucosidase (bglX) and α-glucosidase maltase (malL), but no p-values ​​or statistical tests are mentioned. How was this conclusion reached?
  • Authors are encouraged to rewrite the conclusion to highlight the novelty of the study.

Comments on the Quality of English Language

Language recommendations were made in the observations.

Author Response

Comments 1: Modify title “Effect of Glycosidase Produced by Rhodotorula mucilaginosa on the Release of Flavor Compounds in Fermented White Radish.” By “Effect of Glycosidase Production by Rhodotorula mucilaginosa on the Release of Flavor Compounds in Fermented White Radish.”

Response 1: Thanks for your comments. It has been revised in the current manuscript.

Comments 2: Modify in abstract

"Fermented vegetables are widely favored by consumers due to their unique flavors and rich nutrient profiles. By “Fermented vegetables are highly valued by consumers for their distinct flavors and rich nutritional content.”

"However, there are limited reports on the influence of red yeast on flavor development in fermented vegetables." By “However, studies on the impact of red yeast on flavor development in fermented vegetables remain scarce.”

"When vegetable juice was treated with commercial glycosidase, elevated levels of cis-2-pentenol, hyacinthin, geranylacetone, and 1-dodecanol were detected, mirroring the results obtained from yeast-fermented vegetable juice." By “The application of commercial glycosidase to vegetable juice resulted in increased levels of cis-2-pentenol, hyacinthin, geranylacetone, and 1-dodecanol, consistent with findings from yeast-fermented vegetable juice.” 

Response 2: Thanks for your comments. These sentences have been revised as suggested, please check Line (14,17,26).

Comments 3: Modify in introduction

"Fermented vegetables are traditional foods made to extend the shelf life of vegetables and are widely favored by consumers for their unique flavors." By “Fermented vegetables are traditional foods that extend the shelf life of vegetables and are appreciated for their unique flavors.” 

Response 3: Thanks for your suggestion. It has been revised in the current manuscript.

Comments 4: Review the document and abbreviate minutes as min; In page 4 line 154 separate 72h as 72 h; Check throughout the document that the degrees Celsius are separated from the number (e.g. 32 °C);Check throughout the document that the % symbol is separated from the number (e.g. 32 %). 

Response 4: Thanks for your suggestions. We have carefully revised the manuscript according to your recommendations regarding unit formatting.

Comments 5: In the preparation of the fermentation medium, the authors do not clearly specify the controls used. It is recommended that they be added. 

Response 5: Thank you for your insightful comments on our manuscript. The preparation of control medium has been added in the new revision of this manuscript:

The blank control medium is yeast extract peptone dextrose (YPD) medium, which consists of 20 g/L glucose, 20 g/L peptone, and 10 g/L yeast extract.[Line 122-123]

Comments 6: Why was 600 MPa chosen for high pressure processing (HPP)? 

Response 6: Thank you for your careful review and valuable comments on our manuscript. According to our previous research, using this parameter for high-pressure processing of vegetable juice can accelerate the clarification and filtration of the juice. Therefore, this treatment condition was continued in this study.

References: 

  1. Wang, R., Zeng, Y., Liang, J., Zhang, H., Yi, J., & Liu, Z. (2024). Effect of Rhodotorula mucilaginosa inoculation on the aroma development of a fermented vegetables simulated system. Food Research International, 179. https://doi.org/10.1016/j.foodres.2024.113941

Comments 7: It is claimed that yeast significantly increased the levels of β-glucosidase (bglX) and α-glucosidase maltase (malL), but no p-values ​​or statistical tests are mentioned. How was this conclusion reached? 

Response 7: Thanks for your comments. As shown in Figure A2, we can observe that after fermentation of I2002 in YPD medium, the activities of three glycosidases were significantly enhanced. As shown in Figure A2, the differential gene expression levels of glycosidases in samples from the logarithmic phase and the stationary phase indicate that with the increase in fermentation time, the expression levels of three glucosidase genes annotated in the KEGG pathway were significantly upregulated.

Comments 8: Authors are encouraged to rewrite the conclusion to highlight the novelty of the study. 

Response 8: Thank you for your insightful comments and suggestions. We have carefully revised the conclusion section to highlight the novelty of our study. Below is the revised conclusion:

The present study offers a novel exploration into the impact of glycosidases produced by the red yeast Rhodotorula mucilaginosa on the release of bound flavor compounds in fermented white radish. Through whole-genome sequencing, a total of 37 glycosidase genes were identified within R. mucilaginosa, with nine corresponding coding genes being annotated. Utilizing the PNPG colorimetric method, the presence of α-galactosidase, β-glucosidase, and α-mannosidase activities in the yeast strain was confirmed. A significant finding of this study is the upregulation of the β-glucosidase gene (bglX) and the α-glucosidase maltase gene (malL) following the inoculation of R. mucilaginosa into fermented radish juice. This upregulation was accompanied by an increase in the content of specific flavor compounds, indicating that these glycosidases are actively involved in the hydrolysis and release of bound flavor compounds in vegetables.

Furthermore, four flavor substances—cis-2-pentenol, hyacinthin, geranylacetone, and 1-dodecanol, were identified in radish juice that had been treated with both red yeast and commercial glycosidase. These glycosides were isolated at various stages of the fermentation process. An analysis of both the expression levels of the glycosidase genes and the concentrations of the flavor substances revealed a positive correlation, wherein the content of the flavor substances rose in tandem with the increased expression of the glycosidase genes.

The novelty of this study lies in the use of multi-omics approaches to comprehensively analyze the role of glycosidases produced by R. mucilaginosa in the release of flavor compounds during vegetable fermentation. This comprehensive analysis provides new insights into the formation mechanism of flavor compounds in fermented vegetables and offers a theoretical basis for improving the quality of fermented vegetable products through targeted manipulation of glycosidase activities [Line 473-496].

Round 2

Reviewer 2 Report

Comments and Suggestions for Authors

No comments

Comments on the Quality of English Language

No comments

Author Response

Comments 1: Line 100, 163: Centrifugation units should be in g, not rpm, or you must include rotor information. In section 2.1., also include information about the temperature of centrifugation (similar to section 2.2).

Response 1: Thank you for highlighting this oversight. We have carefully revised the centrifugation parameters to comply with standard scientific reporting practices. [The location of this change can be found in the revised manuscript - lines 100 and 163.]

 Comments 2: Although it is an editorial work, according to authors instructions, References must be numbered in order of appearance in the text (including table captions and figure legends), not alfabetically, and listed individually at the end of the manuscript. In the text, reference numbers should be placed in square brackets [ ], not as (author et al., year).

Response 2: Thank you for highlighting this requirement. We have thoroughly revised the manuscript to ensure full compliance with the reference formatting guidelines.

Comments 3: Also according to authors instructions, all figures, schemes and tables should be inserted into the main text close to their first citation, not at the end of the manuscript. Please, revise authors instructions for a better arrangement of the manuscript.

Response 3: Thank you for emphasizing this critical formatting requirement. We have thoroughly restructured the manuscript to ensure all figures, tables, and schemes are embedded within the main text at their first citation points.

Comments 4: Data Availability Statement has not been included in the manuscript. Please, include it.

Response 4: Thank you for noting this omission. We have added a comprehensive Data Availability Statement to the manuscript in accordance with the journal’s guidelines.[The location of this change can be found in the revised manuscript - line 536.]
